# Pre-Operative Imaging and Pathological Diagnosis of Localized High-Grade Pancreatic Intra-Epithelial Neoplasia without Invasive Carcinoma

**DOI:** 10.3390/cancers13050945

**Published:** 2021-02-24

**Authors:** Ryota Sagami, Kentaro Yamao, Jun Nakahodo, Ryuki Minami, Masakatsu Tsurusaki, Kazunari Murakami, Yuji Amano

**Affiliations:** 1Department of Gastroenterology, Oita San-ai Medical Center, 1213 Oaza Ichi, Oita, Oita 870-1151, Japan; 2Pancreatic Cancer Research for Secure Salvage Young Investigators (PASSYON), Osaka-Sayama, Osaka 589-8511, Japan; yamaken_volvo@yahoo.co.jp (K.Y.); j-nakahodo@cick.jp (J.N.); ryuki12@kuhp.kyoto-u.ac.jp (R.M.); 3Department of Gastroenterology and Hepatology, Kindai University, Osaka-Sayama, Osaka 589-8511, Japan; 4Department of Gastroenterology Tokyo Metropolitan Cancer and Infectious Disease Center Komagome Hospital, 3-18-22 Honkomagome, Bunkyo-ku, Tokyo 113-8677, Japan; 5Department of Gastroenterology, Tenri Hospital, 200 Mishimacho, Tenri, Nara 632-0015, Japan; 6Department of Diagnostic Radiology, Kindai University Faculty of Medicine, Osaka-Sayama, Osaka 589-8511, Japan; mtsuru@dk2.so-net.ne.jp; 7Department of Gastroenterology, Faculty of Medicine, Oita University, 1-1 Idaigaoka, Hasamacho, Yufu, Oita 879-5593, Japan; murakam@oita-u.ac.jp; 8Department of Endoscopy, Urawa Kyosai Hospital, 3-15-31 Harayama, Midoriku, Saitama 336-0931, Japan; amanoyj@gmail.com

**Keywords:** endoscopic ultrasound, high-grade, intraductal papillary mucinous neoplasm, pancreatic cancer, pancreatic duct, pancreatic ductal adenocarcinoma, pancreatic intra-epithelial neoplasia, pancreatic juice cytology, parenchymal atrophy

## Abstract

**Simple Summary:**

Pancreatic ductal adenocarcinoma (PDAC) is typically associated with an extremely poor prognosis; however, small PDAC tumors show good prognosis. High-grade pancreatic intra-epithelial neoplasia (PanIN), which precedes invasive PDAC, is a primary target for improving the prognosis of PDAC. However, detection of high-grade PanIN without invasive carcinoma by existing imaging modalities is difficult because the lesions are only microscopically detectable. Recent studies have reported the characteristics of imaging findings associated with localized high-grade PanIN and the usefulness of serial pancreatic-juice aspiration cytologic examination as a method to confirm the pre-operative histopathology. In this review, we aimed to clarify recent clinical findings regarding detection of localized high-grade PanIN, which may contribute to improvement of the prognosis of patients with PDAC.

**Abstract:**

Pancreatic ductal adenocarcinoma (PDAC) arises from precursor lesions, such as pancreatic intra-epithelial neoplasia (PanIN) and intraductal papillary mucinous neoplasm (IPMN). The prognosis of high-grade precancerous lesions, including high-grade PanIN and high-grade IPMN, without invasive carcinoma is good, despite the overall poor prognosis of PDAC. High-grade PanIN, as a lesion preceding invasive PDAC, is therefore a primary target for intervention. However, detection of localized high-grade PanIN is difficult when using standard radiological approaches. Therefore, most studies of high-grade PanIN have been conducted using specimens that harbor invasive PDAC. Recently, imaging characteristics of high-grade PanIN have been revealed. Obstruction of the pancreatic duct due to high-grade PanIN may induce a loss of acinar cells replaced by fibrosis and lobular parenchymal atrophy. These changes and additional inflammation around the branch pancreatic ducts (BPDs) result in main pancreatic duct (MPD) stenosis, dilation, retention cysts (BPD dilation), focal pancreatic parenchymal atrophy, and/or hypoechoic changes around the MPD. These indirect imaging findings have become important clues for localized, high-grade PanIN detection. To obtain pre-operative histopathological confirmation of suspected cases, serial pancreatic-juice aspiration cytologic examination is effective. In this review, we outline current knowledge on imaging characteristics of high-grade PanIN.

## 1. Introduction

The incidence of pancreatic cancer (PC) has been increasing worldwide, and PC has the seventh most frequent cause of cancer-related mortality, with more than 432,000 deaths per year [1]. The annual number of deaths caused by PC is more than 33,000 in Japan [2] and more than 45,000 in the United States of America (USA) [3]. In addition, PC is the fourth leading cause of cancer-related death in Japan [2] and the third leading cause of cancer-related death in the USA [3]. However, the incidence of PC is predicted to increase, and PC is expected to become the second leading cause of cancer-related death in the USA by 2030 [4]. Pancreatic ductal adenocarcinoma (PDAC) is the most common type of PC, comprising 90% of all malignant pancreatic neoplasms, and is associated with an extremely poor prognosis [3,4], with a 5-year overall survival rate of less than 10% [3,4,5]. By contrast, the 5-year survival rate of patients with invasive cancers measuring 10 mm or less in diameter is over 80% [6]. Therefore, earlier detection of smaller tumors could improve the prognosis of patients with PDAC. Further improvement of the prognosis of patients with PDAC will require the diagnosis and management of precursor lesions [7]. 

PDAC arises from noninvasive precursor lesions, including noncystic lesions (e.g., pancreatic intra-epithelial neoplasia [PanIN]) and cystic lesions (e.g., intraductal papillary mucinous neoplasm [IPMN] and mucinous cystic neoplasm [MCN]) [8,9]. The 5-year survival rate of patients with high-grade precancerous lesions (mainly high-grade PanIN and IPMN with high-grade dysplasia without accompanying invasive carcinoma; Union for the International Cancer Control stage 0 PC [carcinoma in situ]) [8,10] is over 85%. Therefore, the clinical approach for detecting these lesions in the curable stage is crucial for improving outcomes, and high-grade precancerous lesions are ideal target lesions for early PC detection because of their good prognosis [6,11,12,13]. However, the incidence of high-grade precancerous lesions is only 1.7% among all PDACs [6]. Among high-grade precancerous lesions, high-grade PanIN without invasive carcinoma is regarded as the lesion immediately preceding invasive PDAC, suggesting that this type of PanIN may be a primary target for intervention [7,14]. 

Regrettably, high-grade PanIN is difficult to detect using standard clinical and radiological approaches [7,15] because high-grade PanIN lesions do not form masses and are only histologically identified as microscopic mucinous pancreatic ductal lesions, whereas IPMNs and MCNs are clinically diagnosed as cysts by standard radiological examination [9]. Therefore, most high-grade PanIN analysis has been conducted using surgical specimens that harbor invasive PDAC or autopsy specimens [16,17,18,19]. In clinical practice, high-grade PanIN is difficult to detect without an associated invasive PDAC. In addition, high-grade PanIN is also rarely found in large-scale screening studies of patients with a strongly related family history or with PDAC-associated gene mutations [20,21,22,23,24,25,26,27,28]. 

In recent studies, many cases of high-grade PanIN without invasive carcinoma have been assessed, and the imaging and pathological characteristics of these lesions have gradually become clearer [29,30,31,32,33]. In this article, we review recent developments in our understanding of high-grade PanIN in clinical practice in order to resolve clinical questions regarding the imaging characteristics of high-grade PanIN and to provide perspectives for some future issues.

## 2. Pathological and Genetic Characteristics of PanIN

### 2.1. Pathological Features

Pathologists have recognized precursor lesions in the pancreas for more than 100 years [34]; however, the concept of PanIN was first proposed in 1994 [35]. PanIN is microscopically identified as mucinous lesions in the branches of the pancreatic ducts [8,9] and is characterized by cuboid to columnar cells with varying amounts of apical cytoplasmic mucin and varying degrees of cytologic and architectural atypia. The degree of epithelial atypia can be categorized as PanIN-1, PanIN-2, or PanIN-3. PanIN-1 lesions are characterized by minimal nuclear atypia [8,36]. Alternatively, the World Health Organization (WHO) classification defines PanIN based on a two-tiered grading system into low-grade PanIN (formerly PanIN-1 and -2; Figure 1A) and high-grade PanIN (PanIN-3; Figure 1B) [8,10,36]. Clinical studies have shown that PanIN with low-grade dysplasia is related to a low risk of malignant progression and allows clinical observation. By contrast, high-grade dysplasia is related to a high risk of progression into invasive carcinoma and requires surgical management [8,10,36].

### 2.2. Genetic Features

High-grade PanIN typically exhibits loss of polarity, irregular nuclear stratification, budding of cell clusters into the lumen, severe cytological atypia, and mitoses [8,9,10]. PanIN shows multistep progression from low-grade PanIN (PanIN-1, -2) to high-grade PanIN (PanIN-3) and invasive PDAC, similar to the stepwise carcinogenesis of colorectal carcinoma [8,36]. In this process, *KRAS* mutation plays important roles in early stages of PanIN development and is observed in over 90% of all PanIN cases [37,38,39,40]. Loss of *CDKN2A/p16* expression typically occurs after *KRAS* mutation and is more prevalent in high-grade PanIN (71–83%) compared with low-grade PanIN (32%) [15,41,42]. Loss of expression of *TP53* and *SMAD4* is almost exclusively found in high-grade PanIN [40,42,43]. However, abnormal p53 expression is observed by immunohistochemistry in approximately 20% of isolated high-grade PanIN cases without invasive PC, and intact *SMAD4/DPC4* expression is detected in all high-grade PanIN [44]. Targeted next-generation sequencing of high-grade PanIN lesions without invasive carcinoma identified *KRAS* mutations in 94% of lesions. Notably, *TP53* mutations are rare in high-grade PanIN, and no nonsynonymous mutations in *SMAD4* are detected. Adjacent low-grade PanIN harbors *KRAS* mutations in 94% of lesions; however, mutations in *CDKN2A*, *TP53*, and *SMAD4* have not been identified. These results suggest that inactivation of *TP53* and *SMAD4* may be late genetic alterations, predominantly occurring in invasive PDAC [45]. These discrepancies may be due to the presence or absence of accompanying invasive carcinoma; further analysis is needed.

MicroRNA (miRNA) is also important for the diagnosis of chronic pancreatitis, PanIN, and PDAC because abnormal expression of different miRNAs can be found in pancreatic lesions [46,47]. In addition, many miRNAs show aberrant expression in PanIN lesions and are likely to be important in the development of PDAC [46,48]. In the diagnosis of PanIN, 35 of 700 mRNAs showed altered expression using quantitative real-time polymerase chain reaction. In particular, miR-196b, whose expression is limited to high-grade PanIN or pancreatic cancers, is believed to be useful as a diagnostic biomarker [48]. Further studies of the methods of serum or pancreatic juice miRNA analysis in the diagnosis of high-grade PanIN are required [49,50,51].

## 3. Pancreatic Diseases Associated with High-Grade PanIN 

### 3.1. Pancreatic Ductal Adenocarcinoma

PanIN is more common in cases with invasive PDAC than in those without PDAC [36]. High-grade PanIN is diagnosed in 16–75% of patients with invasive PC [16,17,18,19,42,52,53,54,55,56,57], whereas no high-grade PanIN has been reported in patients with a normal pancreas [12,58]. In particular, high-grade PanIN is also more common in familial PDAC compared with sporadic PDAC [59]. High-grade PanIN coexisting with resected PDAC is also associated with recurrence of PC after surgical resection [60]. Thus, high-grade PanIN appears to be strongly associated with PDAC.

### 3.2. Chronic Pancreatitis

Because chronic pancreatitis is a major risk factor of PDAC [61,62,63,64], high-grade PanIN is detected in only 1.5–4% of patients with chronic pancreatitis [12,41,65]. A cohort study also reported that the incidence of coexisting high-grade PanIN with autoimmune pancreatitis is 3.6% [66]. Gradual increases in the rates of coexisting high-grade PanIN in patients with normal pancreas (0%), chronic pancreatitis (4%), and PDAC (40%) demonstrate the close relationship between high-grade PanIN and inflammation associated with chronic pancreatitis during the carcinogenesis of PDAC [12]. In addition, high-grade PanIN concomitant with chronic pancreatitis is reported to be a major prognostic factor affecting the poor survival rates of patients with PDAC [67]. Thus, chronic inflammation may accelerate the progression of pre-invasive high-grade PanIN, similar to dysplasia-associated lesions in chronic ulcerative colitis [67,68], and the coexistence of high-grade PanIN should be suspected in cases with chronic pancreatitis, including autoimmune pancreatitis.

### 3.3. IPMN

IPMN is a cystic neoplasm that communicates with the pancreatic ductal system and a precursor lesions of PDAC [9,69,70,71]. IPMN can be categorized as main-duct or branch-duct type based on the location of the involved pancreatic duct and the presence of cystic dilation of branch ducts [72,73]. Some guidelines recommend surgical resection for main-duct IPMN owing to the high malignant potential [72,73,74]. In contrast, among branch-duct IPMN, the incidence rate of pancreatic carcinogenesis during the follow-up period has been reported to be approximately 3–9.3% [71,75,76,77,78,79]; therefore, careful surveillance is needed. In addition, surveillance of branch-duct IPMN should be focused on two types of carcinogenesis, carcinoma derived from IPMN; intraductal papillary mucinous carcinoma (IPMC) and concomitant PDAC (de novo PDAC) [78,79,80,81,82]. 

#### 3.3.1. IPMN with High-Grade Dysplasia

IPMN exhibits multistep progression of low-grade IPMNs to high-grade IPMN and IPMC [9]. The American Gastroenterological Association and the International Association of Pancreatology have described the high-risk radiological futures of IPMN [72,83], and the resection of branch-duct IPMN is performed based on these guidelines [72,73,82]. Mural nodules may be the most predictive finding of high-grade IPMN and IPMC [83,84,85] and can be detected by existing radiological imaging modalities, such as enhanced computed tomography (CT), magnetic resonance imaging (MRI), and endoscopic ultrasound (EUS) [73,86,87,88]. The diagnostic sensitivity and specificity of mural nodules 5–10 mm in diameter in the context of IPMC and high-grade IPMN are 73–100% and 73–85%, respectively [89,90,91,92,93,94,95]. The cutoff size for mural nodules is 5 mm or more [96], and most branch-duct IPMNs without mural nodules remain unchanged during long-term follow-up (median, 57 months) [85]. Pancreatoscopy is also useful because it enables direct visualization of lesions in pancreatic duct and direct biopsy [97]. In particular, this approach is useful for differential diagnosis between malignant and benign IPMN with an accuracy of 67% for branch duct IPMNs [98]. Using the technique of EUS-fine needle aspiration (EUS-FNA), cyst fluid analysis and confocal laser endoscopy can be performed [7,88]. Cyst fluid cytology has 90% specificity for the diagnosis of high-grade IPMN and IPMC [66,88,99]. In addition, to detect small IPMC or high-grade IPMN, DNA-based examination of pancreatic cyst fluid is useful [100], showing a sensitivity and specificity of 89% and 100%, respectively, which are higher than those of mural nodules (32% and 94%, respectively) and malignant cytopathology (32% and 98%, respectively) [101]. Confocal laser endoscopy identifying a vascular network pattern representing subepithelial capillary vascularization using endomicroscopy may useful for distinguishing high-grade IPMN and IPMC from IPMN without high-grade dysplasia, with 83–88% sensitivity and 88–100% specificity [102,103]. Thus, imaging and histopathological diagnostic methods for IPMC or high-grade IPMN have been established to some extent.

#### 3.3.2. High-Grade PanIN Associated with IPMN

Concomitant PDAC, which is found in approximately 40% of detected PDAC with branch-duct IPMN at first examination or during follow-up, does not occur from IPMN lesions themselves [78,82]. PanIN lesions, particularly high-grade PanIN lesions found in pancreatic ducts unrelated to IPMN, should be associated with the development of concomitant invasive PDAC coexisting with IPMN [104]. High-grade PanIN is detected in 6.3–19% of resected pancreatic specimens of branch-duct IPMNs [55,105,106]. Additionally, another study reported that high-grade PanIN is detected in 2.4% of cases with multiple small cystic pancreatic lesions, which may be branch-duct type IPMNs in patients with familial cancer [107]. In addition, IPMNs coexisted in 39.2% of resected cases of high-grade PanIN without invasive carcinoma [29]. Therefore, high-grade PanIN coexisting with IPMN is also an ideal target lesion for early detection of PDAC. However, imaging diagnostic method of high-grade PanIN coexisting IPMN has not been established compared to high-grade IPMN itself.

### 3.4. Other Pancreatic Diseases

Another study retrospectively reviewed patients with PanIN who underwent pancreatectomy for non-IPMN and non-PDAC (e.g., neuroendocrine tumor, serous cystadenoma, mucinous cystic neoplasm, and solid pseudopapillary tumor); the findings showed that 2.2% of patients had high-grade PanIN lesions [108]. High-grade PanIN lesions rarely coexist with pancreatic lesions to be resected other than IPMN and PDAC. 

### 3.5. Aging Pancreas

The frequency of PanIN increases with age [12,36,58]. Aging is also thought to promote a sequential change from nonpapillary hyperplasia to papillary hyperplasia, atypical hyperplasia, and finally to invasive carcinoma [52]. High-grade PanIN is detected in 4% of autopsy cases (mean age, 80.5 years), with no evidence of PDAC and/or IPMN [19]. The presence of high-grade PanIN lesions can be clarified by pancreatic autopsy specimens without clinical detection prior to autopsy. In elderly patients, age-related pathological changes, such as PanIN, fatty replacement, lobulocentric pancreatic atrophy, and pancreatic duct dilation, are thought to play key roles in pancreatic carcinogenesis [109]. Generally, particularly in elderly patients with IPMN, chronic pancreatitis, and other pancreatic tumors, co-existence of high-grade PanIN should be monitored carefully.

## 4. Imaging Characteristics of High-Grade PanIN

### 4.1. Relationship between Imaging Findings and Pathological Features Associated with of PanIN

Most high-grade PanIN without invasive carcinoma cannot be detected as obvious findings by any imaging modality because the intra-epithelial changes of the lesions are microscopic [9]. Therefore, indirect imaging findings may play key roles in the detection of high-grade PanIN. Indirect imaging findings associated with high-grade PanIN may be considered from a viewpoint of histopathological characteristics. PanIN is a type of multifocal lesion and is more commonly located in the head of the pancreas than in the body or tail [12,52]. Because more than half of PDAC tumors occur in the pancreatic head [9], the incidence of dysplasia in the head is higher than that in the tail, providing a potential explanation for the preferred localization of PDAC in the pancreatic head. Notably, PanIN develops within the micro-cystically dilated intralobular glands and ducts [110]. Narrowing or obstruction of the pancreatic duct may induce a thinning of the acinar cells and obstruction of secretion. The thinning of the acinar cell layer is followed by apoptosis and/or necrosis-mediated cell death. Loss of acinar cells is replaced by fibrotic changes and infiltration of immune cells, and lobulo-centric parenchymal atrophy is induced [58,109,111]. In addition, these changes may accompany the multistep progression of PanIN [112]. Around the branch pancreatic ducts (BPDs), obstructive lobular changes, parenchymal atrophy, fibrosis, and inflammatory changes that are induced by multifocal PanIN may cause main pancreatic duct (MPD) stenosis, MPD dilation, multiple microcystic lesions (BPD dilation), and heterogeneous parenchymal atrophy, such as chronic pancreatitis. [58,113,114,115]. Representative histopathological and imaging findings of pancreatic parenchymal changes around high-grade PanIN are shown in Figure 2A. Even if PanIN is too small to visualize by existing imaging modalities, the additional findings induced by the pancreatic parenchyma and fibrosis, such as morphological changes in the ducts, microcystic changes, and localized parenchymal atrophy, can be detected and quantified by existing modalities. 

### 4.2. Indirect Imaging Characteristics of High-Grade PanIN

To detect the localization of high-grade PanIN before resection, indirect imaging findings should consider the pathological features associated with PanIN, because high-grade PanIN without formation of a definite mass cannot be detected by any imaging modality. A Japanese nationwide study of early pancreatic cancer reported that stage 0 PC, including only high-grade PanIN without invasive carcinoma but excluding high-grade IPMN, could be diagnosed pre-operatively based on indirect imaging findings, such as MPD dilation, MPD stenosis, and focal pancreatic parenchymal atrophy (PPA) [29]. In addition, hypoechoic changes and retention cysts (BPD dilation) around MPD stenosis that can be diagnosed by EUS are also candidates for accompanying imaging findings of high-grade PanIN [30,31]. According to these concepts, in Japan, many cases of localized high-grade PanIN may be diagnosed and treated successfully. 

#### 4.2.1. Morphological Changes in the Main Pancreatic Duct (Stenosis and Dilation)

MPD dilation is a predictor of PDAC, and localized MPD stenosis with or without upstream MPD dilation is often observed in PDAC. The detection and follow-up of these MPD findings are necessary to diagnose PDAC and high-grade PanIN without invasive carcinoma [116,117,118]. Abrupt MPD stenosis is a significant factor in patients with small PDAC because stenosis in patients with benign pancreatic diseases is generally slow growing [119,120,121,122]. In addition, the long segment of MPD stenosis and non-MPD penetration are also important in patients with PDAC, in contrast to those in patients with benign MPD stenosis [123]. Generally, MPD dilation is thought to be caused by pancreatic juice flow obstruction owing to downstream pancreatic duct stenosis, and any type of stenosis with upstream MPD dilation could support the need for further examinations to detect high-grade PanIN [31]. A representative case of MPD dilation and stenosis with high-grade PanIN is shown with schema in Figure 2B–E.

The rate of MPD dilation is reported to be 72–83% by various imaging modalities, including ultrasonography (US), CT, MRI, EUS, and endoscopic retrograde cholangiopancreatography (ERCP), in a retrospective multicenter study summarizing a relatively large number of 51 high-grade PanIN cases [29]. Additionally, that of MPD stenosis is reported to be 68% (28/41) by EUS and 83% (39/47) by ERCP. In another 27 high-grade PanIN cases, the rate of MPD dilation is reported to be 44–52% by CT and MRI, and the rate of MPD stenosis is reported to be 74% (20/27) by EUS and 44% by ERCP [33]. In pre-operative imaging of resected high-grade PanIN cases, MPD dilation and stenosis are detected in 44–100% and 6–100% of cases, respectively, (Table 1) [29,30,31,32,33]. Overall, ERCP more clearly clarified the irregularities in MPD caused by the disappearance of BPDs [33]. By contrast, in 50% (3/6) of cases of high-grade PanIN, only MPD dilation without downstream stenosis is detected [31]. In such cases, the gradient of intraductal pancreatic juice pressure may cause pancreatic duct dilation as PPA progressed around high-grade PanIN [31]. Although localized MPD stenosis or dilation is also observed in patients with other benign pancreatic diseases, such as chronic pancreatitis, including autoimmune pancreatitis, unnatural MPD dilation with and without downstream stenosis should be assessed for detection of high-grade PanIN [123].

#### 4.2.2. Retention Cysts (Dilation of BPDs)

Small cystic lesions, including retention cysts (BPD dilation), are considered a risk factor for PDAC [117,124,125], and long-term follow-up of pancreatic cysts contributes to detection of PDAC and high-grade PanIN [117]. Canto et al. [23] reported the usefulness of retention cyst detection in the diagnosis of curable and noninvasive high-grade neoplasms. Kimura et al. [126] reported that some retention cysts measuring less than 4 mm in diameter are relevant for the diagnosis of high-grade PanIN compared with larger cysts. In some studies, including autopsy cases, approximately 4% of pancreatic cyst cases exhibit duct dilation associated with high-grade PanIN [19,126]. In cases with retention cysts coexisting with PanIN, micro-cystically dilated intralobular ducts, including high-grade PanIN inside and cyst formation (e.g., chronic pancreatitis) induced by multifocal PanIN, may cause retention cysts [58,113,114]. Epithelia of dilated ductal branches, which are sometimes adjacent to cystic lesions, show a similar degree of atypia as those of the cystic lesions themselves [126]. Alternatively, MPD stenosis induced by high-grade PanIN may also cause micro-retention cysts [115]. Overall, retention cysts themselves may include high-grade PanIN or may be indirect findings related to high-grade PanIN lesions. A representative case of retention cyst with high-grade PanIN is shown with schema in Figure 2B,C. In pre-operative imaging of resected cases with high-grade PanIN, 2 studies report retention cysts, and the detection rates are 74% (20/27) by CT/MRI and 31% (5/16) by EUS (Table 1) [30,33]. In these reports, most cases of high-grade PanIN with retention cysts have been identified by morphological changes in MPD. Retention cysts are typical indirect imaging characteristics accompanying high-grade PanIN.

#### 4.2.3. Focal PPA

PPA is an indirect imaging finding associated with PDAC. Some studies have reported that the presence of PPA is an important factor in diagnosis of PDAC [123,127,128,129] and a significant indicator of PDAC rather than autoimmune pancreatitis [129]. Lobular parenchymal atrophy is thought to be induced by lobular fibrosis [33,58,130]. In addition, pancreatic acinar atrophy associated with high-grade PanIN could show dominance of fatty changes or fibrotic changes [33]. Thus, previous pathological studies have supported that PPA is a characteristic pathological finding accompanying high-grade PanIN. Some studies have evaluated the presence of upstream PPA but not focal PPA in patients with PDAC and reported that focal PPA may be highly associated with the presence of small PDAC (≤10 mm) and high-grade PanIN, although upstream PPA is considered an indicator of PDAC [29,32,33,123,127,128,129]. In addition, focal PPA may also be important for distinguishing PPA associated with high-grade PanIN and age-induced diffuse parenchymal atrophy [33]. A representative case of focal PPA with high-grade PanIN is shown in Figure 2C. In pre-operative imaging of cases with resected high-grade PanIN, focal PPA is detected in 42–64% of cases by enhanced CT in 3 studies (Table 1) [29,33,34]. In fact, focal PPA may be a crucial indicator for identification of the localization of high-grade PanIN.

#### 4.2.4. Hypoechoic Changes Around the MPD

The pancreatic parenchyma is usually echogenic owing to refraction of ultrasound by pancreatic acinar, and the lack of pancreatic parenchyma replaced by fibrosis can induce local decreases in echogenicity, resulting in hypoechoic findings [33,44]. Thus, localized inflammation and fibrosis in the interstitial tissue around the pancreatic ducts are detected in hypoechoic areas or masses (such as lesions) by EUS, and the boundary of the surrounding pancreatic parenchyma is consequently unclear. These histopathological changes may occur around high-grade PanIN lesions [30]. Other reports have also suggested that not only fibrosis, but also fatty changes, can contribute to the hypoechoic area around MPD stenosis [31,131]. A representative case of MPD stenosis surrounded by hypoechoic area with high-grade PanIN is shown with schema in Figure 2D. The importance of hypoechoic changes around the MPD in high-grade PanIN cases is reported in two studies, and the detection rates are 74% (20/27) and 56% (9/16), respectively. In pre-operative imaging of cases with resected high-grade PanIN, these hypoechoic changes are detected in 33–74% of cases by EUS only (Table 1) [30,31,33]. Although the objectivity of the findings must be evaluated, hypoechoic areas surrounding the MPD stenosis are useful factors for further examination to detect localized high-grade PanIN.

### 4.3. Recommended Modalities for Detection of High-Grade PanIN

#### 4.3.1. Recommended Modalities

Invasive PDAC is detected as a hypoechoic and hypovascular mass by various imaging modalities. The diagnostic sensitivities of US, CT, MRI, and EUS for the detection of the pancreatic tumors are 67%, 74%, 79%, and 94%, respectively [132]. For detection of relatively small pancreatic tumors measuring approximately about 20 mm in diameter, EUS showed higher sensitivity than enhanced CT (85–94% versus 50–58%, respectively) [133,134]. By contrast, indirect imaging findings should be evaluated in the diagnosis of high-grade PanIN because these lesions are only microscopically detectable and do not form masses. In the detection of pancreatic abnormalities, EUS and MRI play complementary roles, and agreement between EUS and MRI for the detection of clinically relevant lesions is relatively high [23,135]. Two studies comparing high-grade PanIN (one study includes cases of small invasive cancer, with lesions measuring 10 mm or less) and nonmalignant lesions have been published [32,33]. Focal PPA and hypoecho around pancreatic duct stenosis by EUS are significant findings distinguishing high-grade PanIN (and 10 mm PDAC) from nonmalignant lesions; the accuracy, sensitivity, and false positive rate are shown in Table 2.

Considering the detection rate of each modality in previous reports (Table 1 and Table 2) and the invasion of each modality, such as the risk of radiation exposure and complications of endoscopy, EUS and MRI may be suggested as optimal imaging modalities for detection of MPD changes and retention cysts (Table 3). Furthermore, only EUS can detect hypoechoic changes around the MPD, and enhanced CT may be superior for the diagnosis of focal PPA (Table 3). 

#### 4.3.2. Other Newer Imaging Modalities

Contrast-enhanced EUS to evaluate the vascularity of lesions is often critical for the characterization of solid lesions, including PDAC, with a sensitivity and specificity of 88–94% and 88–90%, respectively [132,136,137]. In contrast, those for malignant pancreatic diseases are reported to be 95% and 53%, respectively [138]. EUS elastography to calculate the stiffness of the target tissue is also used to characterize pancreatic masses and lymph node metastases of PDAC, with a sensitivity and specificity of 93–99% and 63–76%, respectively [106,139,140,141,142,143,144,145]. The fibrotic area around high-grade PanIN may be detectable by these imaging methods [146,147]. However, high-grade PanIN usually does not exhibit mass formation, and the case number is low; therefore, the usefulness of these modalities for the diagnosis of high-grade PanIN is unclear. Pancreatoscopy is also useful to directly observe and biopsy lesions in the pancreatic duct [97,98] and can differentiate neoplastic pancreatic changes from those of benign lesions with a sensitivity and specificity of 91% and 95%, respectively [148]. Although pancreatoscopy may be useful in the diagnosis of benign and malignant pancreatic duct changes, such as chronic pancreatitis and PDAC [148,149], there is insufficient evidence regarding the usefulness of this approach for the diagnosis of high-grade PanIN. Notably, pancreatoscopy is associated with the following limitations: high complication rate of pancreatitis (10–12%) and low visualization rate of Wirsung ducts (70–80%). In addition, this method is inappropriate for cases with a main pancreatic duct diameter less than 5 mm [97]; indeed, most cases of high-grade PanIN do not exhibit MPD dilation to that extent. Computerized tools that convert images into quantitative mineable data (radiomics) and subsequent analyses using artificial intelligence may be useful for the diagnosis of PDAC and malignant IPMN [150,151]. Further studies of the diagnosis of high-grade PanIN using these approaches are needed.

### 4.4. Differential Diagnosis Between High-Grade PanIN and Benign Lesions (Combination of Indirect Imaging)

Indirect imaging findings, such as MPD caliber changes and retention cysts, are observed in patients with benign pancreatic diseases, such as chronic pancreatitis and autoimmune pancreatitis [123]. A study of long-term follow-up of patients with MPD stenosis with upstream dilation reports that 47% of cases are diagnosed with PDAC and high-grade PanIN; however, differential diagnosis from benign lesions is difficult [152]. In diagnoses based on findings of MPD stenosis and dilation, a combination of indirect imaging findings may be useful to detect high-grade PanIN or benign disease. The combinations of MPD dilation with focal PPA (44.4% versus 10.5%, *p* = 0.022, specificity: 89.5%), MPD dilation with hypoechoic changes around MPD stenosis (40.7% versus 5.3%, *p* = 0.008, specificity: 94.7%), and focal PPA with hypoechoic changes around stenosis (44.4% versus 10.5%, *p* = 0.022, specificity: 89.5%) have been reported to significantly distinguish between high-grade PanIN and nonmalignant lesions [33].

MPD stenosis with focal PPA and upstream MPD dilation is also considered an important factor for the diagnosis of high-grade PanIN and small PDAC [2]. Among patients with focal PPA corresponding to the distribution of MPD stenosis, upstream PPA arising from the site of MPD stenosis may be also significantly higher in patients with small PDAC (≤10 mm) and high-grade PanIN than in those with nonmalignant MPD stenosis lesions (45.8% versus 7.1%, *p* < 0.01; 33.3% versus 3.6%, *p* = 0.01, respectively) [32]. However, determining the surgical indication from only these indirect imaging findings is still difficult, and histopathological confirmation with pancreatic juice cytology is necessary before surgical resection [29,30,31,32,33]. 

## 5. Pre-Operative Histopathological Diagnosis of High-Grade PanIN

Pre-operative histopathological diagnosis of PDAC is performed using two diagnostic approaches, i.e., EUS-FNA and pancreatic juice cytology. EUS-FNA has been widely used for histopathological confirmation of pancreatic tumors, including small pancreatic tumors measuring 10 mm or less in diameter, with a high sensitivity of 94% [153,154,155,156,157]. However, EUS-FNA is not applicable for the diagnosis of high-grade PanIN because this type of intra-epithelial lesion cannot be visualized using EUS. Therefore, pancreatic juice cytology is an alternative histopathological diagnostic procedure for the diagnosis of high-grade PanIN. Pancreatic juice cytology with intraductal catheter aspiration during ERCP is reported as more useful method compared to pancreatic juice cytology by duodenal aspiration [158], however, the sensitivity of pancreatic juice and brush cytology during ERCP is not high (sensitivity: 31–66%; accuracy: 47–76%) [29,44,159,160,161,162,163]. Serial pancreatic juice aspiration cytologic examination (SPACE), a relatively new diagnostic method using a naso-pancreatic tube placed via the major papilla by ERCP [164], is preformed mainly in Japan for the diagnosis of high-grade PanIN and small PDAC. SPACE may have additional diagnostic effects for single pancreatic juice cytology during ERCP because this method can be used to carry out multiple pure pancreatic juice cytology samplings using a naso-pancreatic tube [118,162]. Overall, SPACE shows a high sensitivity of 33–100% for the detection of high-grade PanIN and small PDAC [44,164,165,166].

In high-grade PanIN cases, the cytological positive rate of SPACE is 72–83% [29,30,31,44], whereas that of brush cytology is 43% [29]. Pancreatic juice examination is also useful when screening for mutant genes shed from PanIN [167,168]. Some studies have demonstrated that PanIN-associated mutations, including *KRAS*, can be detected in pancreatic juice, despite unremarkable pancreatic findings on imaging [169,170,171]. The usefulness of measuring the number and frequency of different mutations, particularly *TP53/SMAD4* mutations, in pancreatic juice to predict the presence of PDAC or high-grade dysplasia has been reported, and pancreatic juice analysis has the potential to complement existing pancreatic imaging examinations and facilitate evaluation of PanIN [170]. 

Pancreatic juice cytology and SPACE may be useful for detection of high-grade PanIN, although post-ERCP pancreatitis, including the SPACE technique, may be the most serious potential adverse event, with an incidence rate of 0–7.5% [118,158,159,164,172,173]. Further studies are needed to clarify the differences in adverse event rates between pancreatic juice cytology during ERCP and SPACE.

In addition, from cytological analysis of pancreatic juices, it is impossible to completely exclude the presence of malignancy prior to the surgical resection. Considering these risks, SPACE should be performed only in cases with strong suspicion of high-grade PanIN based on imaging findings, as described above. Suggested criteria for diagnosis of high-grade PanIN are shown in Figure 3.

## 6. Challenges to Be Solved in Diagnosis of High-Grade PanIN

### 6.1. Populations Requiring Imaging Analysis to Assess High-Grade PanIN

A clear strategy to select patients who should be surveyed for PDAC or high-grade precancerous lesions (high-grade PanIN and high-grade IPMN) is needed because the prevalence of PDAC is low (12.9 cases per 100,000 person-years) [21]. PDAC screening is recommended only for patients with a certain genetic or familial risk of PDAC (high-risk individuals) and is not recommend for the asymptomatic general population with other risk factors, such as diabetes mellitus, because the detection rate of PDAC is low (1.6%), even in patients with increased familial and genetic risk [174]. However, in this review, the detection rate of high-grade precancerous lesion is not mentioned. In another review, the detection rate of high-grade precancerous lesions and invasive PDAC is reported to be 0.74% for high-risk individuals [20]. Additionally, another recent review reports that the detection rate of high-grade precancerous lesions and T1N0M0 PDAC is 0.9% for high-risk individuals [22]. However, the high-grade precancerous lesions reported in these reviews are mainly high-grade IPMNs, and only a few high-grade PanIN cases included. Hanada et al. [92] focused not only on familial or genetic risk but also clinical findings (tumor markers, pancreatitis, pancreatic enzyme, ultrasound findings, and other risk factors) and found a relatively high diagnostic rate for high-grade PanIN and stage 1 PDAC (0.78%). In general, only 25% of patients with high-grade PanIN and early-stage PDACs have symptoms [29], suggesting difficulties in early-stage diagnosis. Thus, a method for high-grade PanIN screening for symptomatic patients and for asymptomatic patients with risk factors should be also established as quickly as possible. Moreover, the efficacy of surveillance for decreasing the morbidity and mortality rates in screened patients with PDAC risk should be also investigated, and the advantages and disadvantages of screening modalities, such as EUS or ERCP and surgical intervention, should be also evaluated with long-term observations [7,21,174].

### 6.2. Challenges in Diagnosis and Follow-Up

Some challenges in the diagnosis of high-grade PanIN must be overcome to improve outcomes. First, most cases of PDAC occur in the pancreatic head; however, more than half of cases of high-grade PanIN occur in the pancreatic body and tail [29,31,33]. High-grade PanIN in the pancreatic body and tail may be more susceptible to fibrosis, PPA, and chronic pancreatitis-like changes than those in the pancreatic head [109,112]. The imaging characteristics of high-grade PanIN in the pancreatic head should be analyzed in greater detail. Second, difficulties in surgical resection should be considered. Based on current knowledge, the decision regarding surgical resection is made according to highly suspicious evidence of localized high-grade PanIN based on abnormal indirect imaging findings and cytological examination. However, determining the range of pancreatic resection is often difficult because high-grade PanIN can occur multifocally [109] and because cells of high-grade PanIN can progress through the pancreatic ductal system [175]. In addition, harmless benign pancreatic lesions can mimic these high-grade precancerous lesions [7,33]. Surgical operations were performed for 31 patients with strongly suspected high-grade PanIN based on indirect imaging findings and cytological results from SPACE, and 2 cases (6.5%) were diagnosed with lesions without high-grade PanIN (low-grade PanIN) [33]. Surgical pancreatic resection, which is associated with a nontrivial risk of mortality, sometimes can be performed to remove these benign lesions. However, surgical resection is sometimes the only way to diagnose suspected pancreatic lesions definitively, particularly in cases of high-grade PanIN. Moreover, the determination whether PanIN lesions are high- or low-grade is impossible without histopathological examination of resected pancreas specimens. The surveillance of remnant pancreas tissue after resection of high-grade PanIN is also controversial [29,82,176]. The rate of recurrence in the remnant pancreas after resection of high-grade PanIN without invasive carcinoma is 0% [108]. However, another study recommends that clinicians perform follow-up every 3–12 months for at least 5 years after surgical resection of early-stage PDAC and high-grade PanIN because of the high recurrence rate of 15.5% in resected cases [29]. Thus, improvement of imaging detection technologies and long follow-up should be considered for surgically resected cases. 

With regard to histopathology, there are some additional challenges to be overcome. For example, most high-grade PanIN lesions are thought to progress immediately to invasive PDAC [12,14]. However, earlier studies have shown that many cases of high-grade PanIN show intraductal spread associated with invasive cancer [177]. High-grade PanIN with PDAC can be classified into flat, mixed, and low-papillary types (18.3%, 34.1%, and 47.6%, respectively) [178]. They also report that the low-papillary type exhibits a greater tendency than the flat type to invade after wide intraductal spread, whereas the flat type appears to invade with little intraductal spread. In addition, high-grade PanIN without invasive carcinoma may be biologically different from high-grade PanIN with associated PDAC because of the low rates of TP53 and SMAD4 mutations in high-grade PanIN without invasive PDAC (11.8% and 0%, respectively). Thus, the characteristics of high-grade PanIN without invasive carcinoma should also be analyzed with further development of genetic and pathological diagnosis of high-grade PanIN, including analyses of subtype and pathways related to progression [42].

The initiation of tumor cells in the pancreas requires an average growth period of over 12 years, and low-grade PanIN is thought to progress to high-grade PanIN in the final few years [179,180]. Thus, there may be insufficient time for the diagnosis of high- grade PanIN without invasive carcinoma.

### 6.3. Limitation of High-Grade PanIN Diagnosis

As described above, detection of indirect imaging findings may be effective for the diagnosis of high-grade PanIN. However, interobserver or intra-observer agreement for imaging findings has not been established. More studies analyzing interobserver diagnostic agreement, e.g., kappa value evaluation, and quantification of findings are needed to confirm intra-observer agreement. Similar problems have been noted in histopathological diagnosis. Although trained pathologists may not have difficulty distinguishing typical low-grade or high-grade PanIN, lesions showing borderline features between low- and high-grade PanIN (previous classification: between PanIN-2 and -3) may be difficult to diagnose, even for experienced pathologists, considering the varied histologic appearance and etiological schemes of the lesions. Indeed, no reports have demonstrated objectivity and agreement among pathologists. In an immunohistochemical analysis of 10 cases of high-grade PanIN, loss of p16 expression is found in five cases (50%), p53 overexpression is found in two cases (20%), and loss of SMAD4 expression is found in no cases (0%) [44]. Variations in immunohistochemical protein expression have been observed in PanIN, and it is therefore difficult to make a diagnosis owing to molecular biological differences because there are few common gene mutations other than KRAS, even for high-grade PanIN [7,9,40,45]. Future studies from histopathological diagnostic methods, such as immunostaining or the use of specific biomarkers to complement the interobserver consensus in histopathological diagnosis of high-grade PanIN, are required. Additionally, further diagnoses based on imaging and histopathological findings may be possible using artificial intelligence. Overall, it will be necessary to accumulate evidence from many studies to overcome the limitations of high-grade PanIN diagnosis. Evaluation of the advantages and disadvantages of invasive pre-operative diagnosis including ERCP, and surgical intervention, is also needed. In addition, most existing studies involve investigation of a small number of cases, therefore, comparative studies of large numbers of cases are needed.

## 7. Conclusions

High-grade PanIN without invasive carcinoma is an ideal target lesion to improve PC prognosis. MPD stenosis, MPD dilation, retention cysts, focal PPA, and hypoechoic changes around MPD are crucial indirect imaging findings of high-grade PanIN which is detectable by existing imaging modalities. These findings should not be missed when determining the localization, which can affect the therapeutic approach. In addition, pancreatic juice cytology, including SPACE, is the only effective method for obtaining a pre-operative histopathological diagnosis of high-grade PanIN. These imaging and histopathological diagnostic methods may contribute to adequate detection of localized high-grade PanIN.

## Figures and Tables

**Figure 1 cancers-13-00945-f001:**
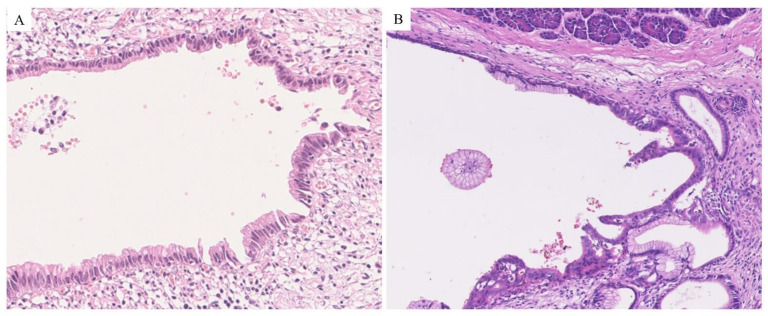
Pathological image of pancreatic intra-epithelial neoplasia (PanIN). (**A**) Low-grade PanIN. Minimal nuclear atypia, absent mitotic figures and flat structure are found. (**B**) High-grade PanIN lesions. Marked atypia consisting of loss of polarity, hyperchromasia, cribriform, micropapillary, and occasional flat architecture are found.

**Figure 2 cancers-13-00945-f002:**
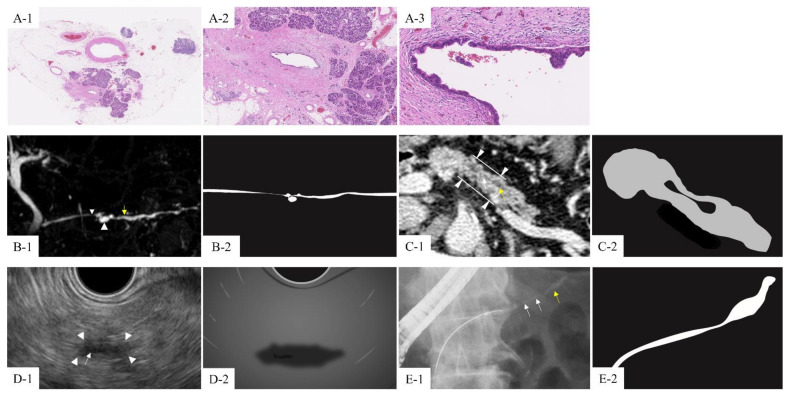
Histopathological and imaging findings of a 73-year-old male with high-grade pancreatic intra-epithelial neoplasia (PanIN). (**A-1**) Weak magnification of the histology of hematoxylin-eosin-stained specimens, showing lobular atrophy and fibrosis around the main pancreatic duct. (**A-2,A-3**) At medium and strong magnification, high-grade PanIN lesions in the lumen show loss of polarity, micropapillary structures, and occasional flat architecture of epithelial cells. Chronic inflammatory cells also infiltrated into the fibrous areas surrounding high-grade PanIN. (**B-1**) Indirect imaging findings associated with high-grade PanIN in MRCP. Main pancreatic duct stenosis (white arrow) with upstream main pancreatic duct dilation (yellow arrow) is shown. Retention cysts are also detectable around the duct stenosis (white arrow head). (**B-2**) Schema of MRCP; main pancreatic duct stenosis with upstream main pancreatic duct dilation and retention cysts around the duct stenosis. (**C-1**) Contrast-enhanced CT showing focal pancreatic parenchymal atrophy at the lesion with a depressed lesion (the range is indicated by arrowheads at the ventral and dorsal sides). The line reveals the margins of the head and tail sides (white line) of high-grade PanIN around main pancreatic duct stenosis (white arrow), with upstream main pancreatic dilation (yellow arrow). (**C-2**) Schema of CT; focal pancreatic parenchymal atrophy around main pancreatic duct stenosis with upstream main pancreatic dilation. (**D-1**) EUS showing high-grade PanIN as a hypoechoic area surrounding (area indicated by the white arrowheads) main pancreatic duct stenosis (white arrow). (**D-2**) Schema of EUS; a hypoechoic area surrounding main pancreatic duct stenosis. (**E-1**) ERCP showing main pancreatic duct stenosis (white arrows) with upstream main pancreatic duct dilation (yellow arrow). (**E-2**) Schema of ERCP; main pancreatic duct stenosis with upstream main pancreatic duct dilation.

**Figure 3 cancers-13-00945-f003:**
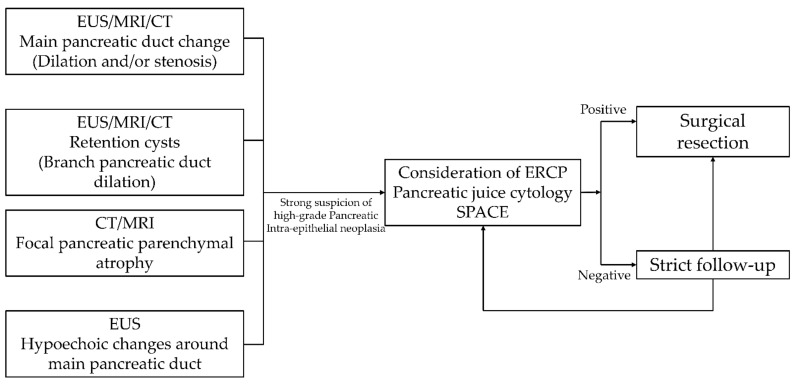
Diagnostic criteria for high-grade PanIN without invasive carcinoma.

**Table 1 cancers-13-00945-t001:** Characteristics of indirect imaging findings of localized high-grade pancreatic intra-epithelial neoplasia without invasive carcinoma.

Study	Total Cases	Imaging Modality	Location Head/Body-tail, *n* (%)	MPD Dilation, *n* (%)	MPD Stenosis, *n* (%)	Retention Cyst, *n* (%)	Focal PPA, *n* (%)	Hypoechoic Changes around MPD, *n* (%)
Yokode [44]	10	CT/MRI/EUS/ERCP	4 (40)/6 (60)	CT 7/10 (70) MRI 7/10 (70) ERCP 4/7 (57)	CT 7/10 (70) MRI 9/10 (90) ERCP 7/7 (100)	NA	NA	3/8 (38)
Kanno [29]	51	US/CT/MRI/ EUS/ERCP	17 (33)/34 (67)	US 26/34 (77) CT 36/50 (72) MRI 34/46 (74) EUS 35/41 (85) ERCP 39/47 (83)	US 2/34 (6) EUS 28/41 (68) ERCP 39/47 (83)	NA	CT 21/50 (42)	NA
Izumi [30]	16	EUS	4 (25)/12 (75)	15/16 (94)	15/16 (94)	5/16 (31)	NA	9/16 (56)
Terada [31]	6	EUS	NA	6/6 (100)	3/6 (50)	NA	NA	2/6 (33)
Yamao [32]	11	CT	NA	NA	NA	NA	7/11 (64)	NA
Nakahodo [33]	27	CT/MRI/EUS/ERCP	5 (19)/22 (82)	CT/MRI 14/27 (52) ERCP 12/27 (44)	EUS 20/27 (74) ERCP 12/27 (44)	CT/MRI 20/27 (74)	CT/MRI 15/27 (56)	20/27 (74)

CT, computed tomography; EUS, endoscopic ultrasound; ERCP, endoscopic retrograde cholangiopancreatography; MPD, main pancreatic duct; MRI, magnetic resonance imaging; NA, not assessed; PPA, pancreatic parenchymal atrophy; US, ultrasonography.

**Table 2 cancers-13-00945-t002:** Accuracy of indirect imaging finding of high-grade PanIN or small invasive cancer in comparison with non-malignant lesion.

Study	Imaging Finding	Accuracy (%)	Sensitivity (%)	Specifisity (%)	False Positive (%)	High-Grade PanIN vs. Non-Malignant Lesion *p* Value
Yamao [32]	Focal PPA (CT)	71	46	93	7	0.004
Nakahodo [33]	MPD dilation (CT, MRI)	57	52	63	37	0.314
	Focal PPA (CT, MRI)	67	56	84	16	0.013
	Retention cyst (CT, MRI)	52	74	21	79	1.000
	Stenosis and hypoecho (EUS)	76	74	79	21	0.001

CT, computed tomography; EUS, endoscopic ultrasound; MPD, main pancreatic duct; MRI, magnetic resonance imaging; PPA, pancreatic parenchymal atrophy.

**Table 3 cancers-13-00945-t003:** Appropriate imaging modalities for detecting indirect imaging findings associated with high-grade pancreatic intra-epithelial neoplasia.

Characteristic of Modalities	MPD Dilation	MPD Stricture	Retention Cyst	Focal PPA	Hypoechoic Changes around MPD
Most sensitive imaging modality	EUS > ERCP > US > MRI > CT	ERCP > EUS / MRI > CT	EUS > MRI > CT > ERCP	CT > MRI	EUS
Invasion of modality	ERCP > EUS > CT > MRI > US	ERCP > EUS > CT > MRI	ERCP > EUS > CT > MRI	CT > MRI	EUS
Appropriate imaging modality	MRI/EUS/US	EUS/MRI	EUS/MRI	CT	EUS

CT, computed tomography; EUS, endoscopic ultrasound; ERCP, endoscopic retrograde cholangiopancreatography; MPD, main pancreatic duct; MRI, magnetic resonance imaging; PPA, pancreatic parenchymal atrophy; US, ultrasonography.

## Data Availability

The dataset used during the current study is available from the corresponding author on reasonable request.

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
