# Peer review of "Pre-Operative Imaging and Pathological Diagnosis of Localized High-Grade Pancreatic Intra-Epithelial Neoplasia without Invasive Carcinoma"

_cancers, 2021, doi:10.3390/cancers13050945_

Round 1

Reviewer 1 Report

The paper "Pre-operative imaging and pathological diagnosis of localized high-grade pancreatic intra-epithelial neoplasia without invasive carcinoma" is an excellent review of the literature.

The identification of high-grade PanIN without invasive carcinoma is of major importance in order to improve the prognosis of PDAC. In this case the uses of standard radiological procedures are often ineffective.

The paper presents very relevant imaging features of high-grade PanIN that represent extremely important elements in its identification and location (high-grade pancreatic intra-epithelial neoplasia without invasive carcinoma).

The paper makes important contributions in terms of imaging and histopathological diagnostic methods in this pathology.

The specialized literature that the authors present is valuable, current and represents important landmarks on the basis of which the conclusions were written.

I mention once again that the work is valuable and has a high scientific level. Suggested improvements are as follows:

Even if the manuscript is not a systematic review, it must still refer to explicit and unitary results identified in several studies. These results must be presented comparatively in order to draw a clear conclusion.   For example, section 4.2 presents “Indirect imaging characteristics of high-grade PanIN”, but no presentation with concrete data on these aspects is made. However in reference [29] ((Kanno, A.; Masamune, A.; Hanada, K.; Maguchi, H.; Shimizu, Y.; Ueki, T.; Hasebe, O.; Ohtsuka, T.; Nakamura, M.; Takenaka, M.; et al. Multicenter study of early pancreatic cancer in Japan. Pancreatology. 2018, 18, 61-67) the results obtained are presented (e.g. see table 3. pp.64 of Kanno et al., 2018).   As the manuscript presents only general, descriptive information, I recommend completing the article, especially sections 4, 5 and 6 with concrete data from the studies considered.

Author Response

February 8, 2021

Prof. Dr. Samuel C. Mok

Editor-in-Chief

Cancers

Manuscript ID: cancers-1093265, R1

Pre-operative imaging and pathological diagnosis of localized high-grade pancreatic intra-epithelial neoplasia without invasive carcinoma

Thank you very much for considering our manuscript and providing us with constructive comments from the Editor and Reviewers. The suggestions have greatly improved our manuscript, and we believe that it is now suitable for publication in Cancers. Point-by-point responses to the Editor’s and Reviewers’ comments are provided in the attached file.

We look forward to hearing from you.

Best regards,

Ryota Sagami, MD

Department of Gastroenterology, Oita San-ai Medical Center

Phone: +81-97-541-5218

Fax: +81-97-541-1311

Reviewer 1.

The paper "Pre-operative imaging and pathological diagnosis of localized high-grade pancreatic intra-epithelial neoplasia without invasive carcinoma" is an excellent review of the literature.

The identification of high-grade PanIN without invasive carcinoma is of major importance in order to improve the prognosis of PDAC. In this case the uses of standard radiological procedures are often ineffective.

The paper presents very relevant imaging features of high-grade PanIN that represent extremely important elements in its identification and location (high-grade pancreatic intra-epithelial neoplasia without invasive carcinoma).

The paper makes important contributions in terms of imaging and histopathological diagnostic methods in this pathology.

The specialized literature that the authors present is valuable, current and represents important landmarks on the basis of which the conclusions were written.

I mention once again that the work is valuable and has a high scientific level. Suggested improvements are as follows:

Even if the manuscript is not a systematic review, it must still refer to explicit and unitary results identified in several studies. These results must be presented comparatively in order to draw a clear conclusion.   For example, section 4.2 presents “Indirect imaging characteristics of high-grade PanIN”, but no presentation with concrete data on these aspects is made. However in reference [29] ((Kanno, A.; Masamune, A.; Hanada, K.; Maguchi, H.; Shimizu, Y.; Ueki, T.; Hasebe, O.; Ohtsuka, T.; Nakamura, M.; Takenaka, M.; et al. Multicenter study of early pancreatic cancer in Japan. Pancreatology. 2018, 18, 61-67) the results obtained are presented (e.g. see table 3. pp.64 of Kanno et al., 2018).   As the manuscript presents only general, descriptive information, I recommend completing the article, especially sections 4, 5 and 6 with concrete data from the studies considered.

Response: We thank the reviewer very much for the constructive comments. According to the Reviewer’s comments, we have added new text or changed the indicated sentences to describe concrete data, as detailed below.

Page 8, lines 308–314

The rate of MPD dilation is reported to be 72–83% by various imaging modalities, in-cluding ultrasonography (US), CT, MRI, EUS, and endoscopic retrograde cholangiopan-creatography (ERCP), in a retrospective multicenter study summarizing a relatively large number of 51 high-grade PanIN cases [29]. Additionally, that of MPD stenosis is reported to be 68% (28/41) by EUS and 83% (39/47) by ERCP. In another 27 high-grade PanIN cases, the rate of MPD dilation is reported to be 44–52% by CT and MRI, and the rate of MPD stenosis is reported to be 74% (20/27) by EUS and 44% by ERCP [33].

Page 8, lines 317–319

By contrast, in 50% (3/6) of cases of high-grade PanIN, only MPD dilation without down-stream stenosis is detected [31].

Page 9, lines 349–351

In pre-operative imaging of resected cases with high-grade PanIN, 2 studies report retention cysts, and the detection rates are 74% (20/27) by CT/MRI and 31% (5/16) by EUS (Table 1) [30,33].

Page 10, lines 367–368

In pre-operative imaging of cases with resected high-grade PanIN, focal PPA was detected in 42–64% of cases by enhanced CT in 3 studies (Table 1) [29,33,34].

Page 10, lines 382–384

The importance of hypoechoic changes around the MPD in high-grade PanIN cases is reported in two studies, and the detection rates are 74% (20/27) and 56% (9/16), respectively.

Page 12, lines 454–461

In diagnoses based on findings of MPD stenosis and dilation, a combination of indirect imaging findings may be useful to detect high-grade PanIN or benign disease. The combinations of MPD dilation with focal PPA (44.4% versus 10.5%, P = 0.022, specificity: 89.5%), MPD dilation with hypoechoic changes around MPD stenosis (40.7% versus 5.3%, P = 0.008, specificity: 94.7%), and focal PPA with hypoechoic changes around stenosis (44.4% versus 10.5%, P = 0.022, specificity: 89.5%) have been reported to significantly distinguish between high-grade PanIN and nonmalignant lesions [33].

Page 12, lines 463–467

Among patients with focal PPA corresponding to the distribution of MPD stenosis, upstream PPA arising from the site of MPD stenosis may be also significantly higher in pa-tients with small PDAC (≤ 10 mm) and high-grade PanIN than in those with nonmalignant MPD stenosis lesions (45.8% versus 7.1%, P < 0.01; 33.3% versus 3.6%, P = 0.01, respectively) [32].

Page 12, line 475

with a high sensitivity of 94% [106,111-115]

Page 12, lines 481–483

however, the sensitivity of pancreatic juice and brush cytology during ERCP is not high (sensitivity: 31–66%; accuracy: 47–76%) [29,44,159-163]. Serial pancreatic juice aspiration cytologic examination (SPACE),

Page 12, line 500 to page 13, line 503

although post-ERCP pancreatitis, including the SPACE technique, may be the most serious potential adverse event, with an incidence rate of 0–7.5% [118,158,159,164,172,173]. Further studies are needed to clarify the differences in adverse event rates between pancreatic juice cytology during ERCP and SPACE.

Page 14, lines 552–555

Surgical operations were performed for 31 patients with strongly suspected high-grade PanIN based on indirect imaging findings and cytological results from SPACE, and 2 cases (6.5%) were diagnosed with lesions without high-grade PanIN (low-grade PanIN) [33].

Page 14, lines 561–565

The rate of recurrence in the remnant pancreas after resection of high-grade PanIN without invasive carcinoma is 0% [108]. However, another study recommends that clinicians perform follow-up every 3–12 months for at least 5 years after surgical resection of early-stage PDAC and high-grade PanIN because of the high recurrence rate of 15.5% in re-sected cases [29].

Page 14, lines 571–573

High-grade PanIN with PDAC can be classified into flat, mixed, and low-papillary types (18.3%, 34.1%, and 47.6%, respectively) [178].

Page 14, line 575–581

In addition, high-grade PanIN without invasive carcinoma may be biologically different from high-grade PanIN with associated PDAC because of the low rates of TP53 and SMAD4 mutations in high-grade PanIN without invasive PDAC (11.8% and 0%, respectively). Thus, the characteristics of high-grade PanIN without invasive carcinoma should also be analyzed with further development of genetic and pathological diagnosis of high-grade PanIN, including analyses of subtype and pathways related to progression [42].

Reviewer 2 Report

This is an interesting review on the PanIN as an early lesion of pancreatic ductal adenocarcinoma.

By definition, PanIN is invisible lesion in conventional imaging modalities, it could be detected indirectly such as p-duct caliber change or small parenchymal change. However most of them are reported even in overt early PDAC not only PanIN.

In spite of extensive review of previous articles, most of them were cases reports or series with selection bias. Without balanced viewpoints on this topic, it could mislead the readership for the concept on PanIN, More studies will be needed on this issue.

Author Response

February 8, 2021

Prof. Dr. Samuel C. Mok

Editor-in-Chief

Cancers

Manuscript ID: cancers-1093265, R1

Pre-operative imaging and pathological diagnosis of localized high-grade pancreatic intra-epithelial neoplasia without invasive carcinoma

Thank you very much for considering our manuscript and providing us with constructive comments from the Editor and Reviewers. The suggestions have greatly improved our manuscript, and we believe that it is now suitable for publication in Cancers. Point-by-point responses to the Editor’s and Reviewers’ comments are provided in the attached file.

We look forward to hearing from you.

Best regards,

Ryota Sagami, MD

Department of Gastroenterology, Oita San-ai Medical Center

Phone: +81-97-541-5218

Fax: +81-97-541-1311

Reviewer 2.

This is an interesting review on the PanIN as an early lesion of pancreatic ductal adenocarcinoma.

By definition, PanIN is invisible lesion in conventional imaging modalities, it could be detected indirectly such as p-duct caliber change or small parenchymal change. However most of them are reported even in overt early PDAC not only PanIN.

In spite of extensive review of previous articles, most of them were cases reports or series with selection bias. Without balanced viewpoints on this topic, it could mislead the readership for the concept on PanIN, More studies will be needed on this issue.

Response: We thank the reviewer very much for the constructive comments. As the reviewer mentioned, previous studies describing high-grade PanIN are case reports or case series with a small number of high-grade PanIN cases (up to about 50 cases). Major reports on imaging of high-grade PanIN were collected for this review. We believe that it may be difficult to obtain a worldwide consensus. However, studies of early pancreatic cancer and high-grade PanIN detection have been limited for decades; therefore, it may be necessary to derive some diagnostic clues from existing data. To avoid misleading the readers, we added a new section describing the limitations of diagnosis of high-grade PanIN, as shown below.

Page 15, lines 586–612

6.3 Limitation of high-grade PanIN diagnosis

As described above, detection of indirect imaging findings may be effective for the diagnosis of high-grade PanIN. However, interobserver or intra-observer agreement for imaging findings has not been established. More studies analyzing interobserver diagnos-tic agreement, e.g., kappa value evaluation, and quantification of findings are needed to confirm intra-observer agreement. Similar problems have been noted in histopathological diagnosis. Although trained pathologists may not have difficulty distinguishing typical low-grade or high-grade PanIN, lesions showing borderline features between low- and high-grade PanIN (previous classification: between PanIN-2 and -3) may be difficult to diagnose, even for experienced pathologists, considering the varied histologic appearance and etiological schemes of the lesions. Indeed, no reports have demonstrated objectivity and agreement among pathologists. In an immunohistochemical analysis of 10 cases of high-grade PanIN, loss of p16 expression is found in five cases (50%), p53 overexpression is found in two cases (20%), and loss of SMAD4 expression is found in no cases (0%) [44]. Variations in immunohistochemical protein expression have been observed in PanIN, and it is therefore difficult to make a diagnosis owing to molecular biological differences because there are few common gene mutations other than KRAS, even for high-grade PanIN [7,9,40,45]. Future studies from histopathological diagnostic methods, such as im-munostaining or the use of specific biomarkers to complement the interobserver consensus in histopathological diagnosis of high-grade PanIN, are required. Additionally, further diagnoses based on imaging and histopathological findings may be possible using artificial intelligence. Overall, it will be necessary to accumulate evidence from many studies to overcome the limitations of high-grade PanIN diagnosis. Evaluation of the advantages and disadvantages of invasive pre-operative diagnosis including ERCP, and surgical intervention, is also needed. In addition, most existing studies involve investigation of a small number of cases, therefore, comparative studies of large numbers of cases are needed.

Reviewer 3 Report

The review is informative and is very well written. The tables and figures are very helpful. The references are appropriate. I just have the following minor comments:

  • In addition to data on sensitivity, the authors should include data on accuracy, specificity, and false positive results, with the different imaging modalities. This should be included in a new table.
  • Also, the authors should comment on the level of interobserver and intra-observer agreement for each of the imaging findings. If the kappa is low, the imaging changes will not be as informative.
  • The authors described the performance of conventional imaging methods for the diagnosis of PAN-IN. How about the performance of newer imaging methods such as AI, radiomics, pancreatoscopy during ERCP, contrast-enhanced EUS, EUS elastography? The authors should include a section on newer imaging technologies
  • Although the authors describe appropriately the indirect imaging findings of PAN-IN, one important consideration is the indication for obtaining the imaging. Can the authors include a brief section as to who needs imaging to assess for PAN-IN? It is clear that imaging screening is not helpful for the general population and is not recommended by societies. Furthermore, some of the findings reported in the review are not very specific, and doing an ERCP and/or surgery might result in more harms than benefits.
  • Can you describe the technique of SPACE in more detail? Is this an old technique or is this still used widely? As an advanced endoscopist, I have never done this. What is the difference of this and simple pancreatic aspiration through a catheter or duodenal aspiration? What are the benefits of this vs. other techniques? What is the risk of pancreatitis of putting a naso-pancreatic tube?
  • Is there a role of micro-DNA and RNA for diagnosis of PAN-IN? How about circulating DNA in blood?
  • Can the authors comment about the limitations on pathologic interpretation of PAN-IN? How about inter-observer agreement? The diagnosis of high-grade PAN-IN with HGD is difficult, and resembles that of Barrett's esophagus with HGD.

Author Response

February 8, 2021

Prof. Dr. Samuel C. Mok

Editor-in-Chief

Cancers

Manuscript ID: cancers-1093265, R1

Pre-operative imaging and pathological diagnosis of localized high-grade pancreatic intra-epithelial neoplasia without invasive carcinoma

Thank you very much for considering our manuscript and providing us with constructive comments from the Editor and Reviewers. The suggestions have greatly improved our manuscript, and we believe that it is now suitable for publication in Cancers. Point-by-point responses to the Editor’s and Reviewers’ comments are provided in the attached file.

We look forward to hearing from you.

Best regards,

Ryota Sagami, MD

Department of Gastroenterology, Oita San-ai Medical Center

Phone: +81-97-541-5218

Fax: +81-97-541-1311

Reviewer 3.

The review is informative and is very well written. The tables and figures are very helpful. The references are appropriate. I just have the following minor comments:

In addition to data on sensitivity, the authors should include data on accuracy, specificity, and false positive results, with the different imaging modalities. This should be included in a new table.

Response: We thank the reviewer very much for the important comments. At the moment, only two studies have comparatively evaluated imaging findings of high-grade PanIN and nonmalignant pancreatic lesions. According to the reviewer’s comment, we added some new sentences to the text and constructed a new table (Table 2).

Page 10, lines 400–405

Two studies comparing high-grade PanIN (one study includes cases of small invasive cancer, with lesions measuring 10 mm or less) and nonmalignant lesions have been published [32, 33]. Focal PPA and hypoecho around pancreatic duct stenosis by EUS are significant findings distinguishing high-grade PanIN (and 10 mm PDAC) from nonmalignant lesions; the accuracy, sensitivity, and false positive rate are shown in Table 2.

Also, the authors should comment on the level of interobserver and intra-observer agreement for each of the imaging findings. If the kappa is low, the imaging changes will not be as informative.

Response: We appreciate this constructive comment from the reviewer. To date, no studies have evaluated interobserver or intra-observer agreement for imaging findings. This should be addressed in future studies. Therefore, we have added some text to discuss the limitations of the field, as detailed below.

Page 15, lines 587–591

As described above, detection of indirect imaging findings may be effective for the diagnosis of high-grade PanIN. However, interobserver or intra-observer agreement for imaging findings has not been established. More studies analyzing interobserver diagnostic agreement, e.g., kappa value evaluation, and quantification of findings are needed to confirm intra-observer agreement.

The authors described the performance of conventional imaging methods for the diagnosis of PAN-IN. How about the performance of newer imaging methods such as AI, radiomics, pancreatoscopy during ERCP, contrast-enhanced EUS, EUS elastography? The authors should include a section on newer imaging technologies.

Response: We thank the reviewer for this constructive comment. According to the reviewer’s comment, we added a new section (section 4.3.2.) describing newer imaging technologies for diagnosis of high-grade PanIN.

Page 11, line 424 to page 12, line 447

4.3.2. Other newer imaging modalities

              Contrast-enhanced EUS to evaluate the vascularity of lesions is often critical for the characterization of solid lesions, including PDAC, with a sensitivity and specificity of 88–94% and 88–90%, respectively [132,136,137]. In contrast, those for malignant pancreatic diseases are reported to be 95% and 53%, respectively [138]. EUS elastography to calculate the stiffness of the target tissue is also used to characterize pancreatic masses and lymph node metastases of PDAC, with a sensitivity and specificity of 93–99% and 63–76%, respectively [106,139-145]. The fibrotic area around high-grade PanIN may be detectable by these imaging methods [146,147]. However, high-grade PanIN usually does not exhibit mass formation, and the case number is low; therefore, the usefulness of these modalities for the diagnosis of high-grade PanIN is unclear. Pancreatoscopy is also useful to directly observe and biopsy lesions in the pancreatic duct [97,98] and can differentiate neoplastic pancreatic changes from those of benign lesions with a sensitivity and specificity of 91% and 95%, respectively [148]. Although pancreatoscopy may be useful in the diagnosis of benign and malignant pancreatic duct changes, such as chronic pancreatitis and PDAC [148,149], there is insufficient evidence regarding the usefulness of this approach for the diagnosis of high-grade PanIN. Notably, pancreatoscopy is associated with the following limitations: high complication rate of pancreatitis (10–12%) and low visualization rate of Wirsung ducts (70–80%). In addition, this method is inappropriate for cases with a main pancreatic duct diameter less than 5 mm [97]; indeed, most cases of high-grade PanIN do not exhibit MPD dilation to that extent. Computerized tools that convert images into quantitative mineable data (radiomics) and subsequent analyses using artificial intelli-gence may be useful for the diagnosis of PDAC and malignant IPMN [150,151]. Further studies of the diagnosis of high-grade PanIN using these approaches are needed.

Although the authors describe appropriately the indirect imaging findings of PAN-IN, one important consideration is the indication for obtaining the imaging. Can the authors include a brief section as to who needs imaging to assess for PAN-IN? It is clear that imaging screening is not helpful for the general population and is not recommended by societies. Furthermore, some of the findings reported in the review are not very specific, and doing an ERCP and/or surgery might result in more harms than benefits.

Response: We thank the reviewer for this comment. Accordingly, we added the following sections (6.1 Populations requiring imaging analysis to assess high-grade PanIN and and 6.3 Limitations of high-grade PanIN diagnosis).

Page 13, lines 513– page 14, lines 537

6.1 Populations requiring imaging analysis to assess high-grade PanIN

A clear strategy to select patients who should be surveyed for PDAC or high-grade precancerous lesions (high-grade PanIN and high-grade IPMN) is needed because the prevalence of PDAC is low (12.9 cases per 100,000 person-years) [21]. PDAC screening is recommended only for patients with a certain genetic or familial risk of PDAC (high-risk individuals) and is not recommend for the asymptomatic general population with other risk factors, such as diabetes mellitus, because the detection rate of PDAC is low (1.6%), even in patients with increased familial and genetic risk [174]. However, in this review, the detection rate of high-grade precancerous lesion is not mentioned. In another review, the detection rate of high-grade precancerous lesions and invasive PDAC is reported to be 0.74% for high-risk individuals [20]. Additionally, another recent review reports that the detection rate of high-grade precancerous lesions and T1N0M0 PDAC is 0.9% for high-risk individuals [22]. However, the high-grade precancerous lesions reported in these reviews are mainly high-grade IPMNs, and only a few high-grade PanIN cases included. Hanada et al. [92] focused not only on familial or genetic risk but also clinical findings (tumor markers, pancreatitis, pancreatic enzyme, ultrasound findings, and other risk factors) and found a relatively high diagnostic rate for high-grade PanIN and stage 1 PDAC (0.78%). In general, only 25% of patients with high-grade PanIN and early-stage PDACs have symptoms [29], suggesting difficulties in early-stage diagnosis. Thus, a method for high-grade PanIN screening for symptomatic patients and for asymptomatic patients with risk factors should be also established as quickly as possible. Moreover, the efficacy of surveillance for decreasing the morbidity and mortality rates in screened patients with PDAC risk should be also investigated, and the advantages and disadvantages of screening modalities, such as EUS or ERCP and surgical intervention, should be also evaluated with long-term observations [7,21,174].

Page 15, lines 608–610

Evaluation of the advantages and disadvantages of invasive pre-operative diagnosis including ERCP, and surgical intervention, is also needed.

Can you describe the technique of SPACE in more detail? Is this an old technique or is this still used widely? As an advanced endoscopist, I have never done this. What is the difference of this and simple pancreatic aspiration through a catheter or duodenal aspiration? What are the benefits of this vs. other techniques? What is the risk of pancreatitis of putting a naso-pancreatic tube?

Response: We appreciate this constructive comment from the reviewer. We have explained SPACE in greater detail in the revised manuscript, as described below.

Page 12, lines 479–489

Pancreatic juice cytology with intraductal catheter aspiration during ERCP is reported as more useful method compared to pancreatic juice cytology by duodenal aspiration [158], however, the sensitivity of pancreatic juice and brush cytology during ERCP is not high (sensitivity: 31–66%; accuracy: 47–76%) [29,44,159-163]. Serial pancreatic juice aspiration cytologic examination (SPACE), a relatively new diagnostic method using a naso-pancreatic tube placed via the major papilla by ERCP [164], is preformed mainly in Japan for the diagnosis of high-grade PanIN and small PDAC. SPACE may have additional diagnostic effects for single pancreatic juice cytology during ERCP because this method can be used to carry out multiple pure pancreatic juice cytology samplings using a naso-pancreatic tube [118,162]. Overall, SPACE shows a high sensitivity of 33–100% for the detection of high-grade PanIN and small PDAC [44,164-166].

Page 13, lines 500– page 13, lines 503

although post-ERCP pancreatitis, including the SPACE technique, may be the most serious potential adverse event, with an incidence rate of 0–7.5% [118,158,159,164,172,173]. Further studies are needed to clarify the differences in adverse event rates between pancreatic juice cytology during ERCP and SPACE.

Is there a role of micro-DNA and RNA for diagnosis of PAN-IN? How about circulating DNA in blood?

Response: We thank the reviewer for the constructive comment. We changed word “molecular” to “genetic” in the subtitles for sections 2 and 2.2 to include the commentary regarding microRNAs in the diagnosis of PanIN.

Page 4, lines 137–145

MicroRNA (miRNA) is also important for the diagnosis of chronic pancreatitis, PanIN, and PDAC because abnormal expression of different miRNAs can be found in pancreatic lesions [46,47]. In addition, many miRNAs show aberrant expression in PanIN lesions and are likely to be important in the development of PDAC [46,48]. In the diagnosis of PanIN, 35 of 700 mRNAs showed altered expression using quantitative realtime polymerase chain reaction. In particular, miR-196b, whose expression is limited to high-grade PanIN or pancreatic cancers, is believed to be useful as a diagnostic biomarker [48]. Further studies of the methods of serum or pancreatic juice miRNA analysis in the diagnosis of high-grade PanIN are required [49-51].

Can the authors comment about the limitations on pathologic interpretation of PAN-IN? How about inter-observer agreement? The diagnosis of high-grade PAN-IN with HGD is difficult, and resembles that of Barrett's esophagus with HGD.

Response: We appreciate this important comment from the reviewer. Accordingly, we added a section describing the limitations of pathological interpretation of PanIN.

Page 15 Line 587-608

As described above, detection of indirect imaging findings may be effective for the diagnosis of high-grade PanIN. However, interobserver or intra-observer agreement for imaging findings has not been established. More studies analyzing interobserver diagnos-tic agreement, e.g., kappa value evaluation, and quantification of findings are needed to confirm intra-observer agreement. Similar problems have been noted in histopathological diagnosis. Although trained pathologists may not have difficulty distinguishing typical low-grade or high-grade PanIN, lesions showing borderline features between low- and high-grade PanIN (previous classification: between PanIN-2 and -3) may be difficult to diagnose, even for experienced pathologists, considering the varied histologic appearance and etiological schemes of the lesions. Indeed, no reports have demonstrated objectivity and agreement among pathologists. In an immunohistochemical analysis of 10 cases of high-grade PanIN, loss of p16 expression is found in five cases (50%), p53 overexpression is found in two cases (20%), and loss of SMAD4 expression is found in no cases (0%) [44]. Variations in immunohistochemical protein expression have been observed in PanIN, and it is therefore difficult to make a diagnosis owing to molecular biological differences because there are few common gene mutations other than KRAS, even for high-grade PanIN [7,9,40,45]. Future studies from histopathological diagnostic methods, such as im-munostaining or the use of specific biomarkers to complement the interobserver consensus in histopathological diagnosis of high-grade PanIN, are required. Additionally, further diagnoses based on imaging and histopathological findings may be possible using artificial intelligence. Overall, it will be necessary to accumulate evidence from many studies to overcome the limitations of high-grade PanIN diagnosis.

Reviewer 4 Report

This is an excellent review highlighting the field of high-risk pancreatic lesions progressing to PDAC. The review extensively addresses multiple aspects of precursor lesions - PanIN and IPMN progressing to PDAC. While the review leans heavily into PanIN and early detection of high-grade PanINs, there is minimal detail about IPMNs with high-grade dysplasia. I recommend that the authors consider adding a paragraph to address investigations in early and/or accurate detection of high-grade dysplasia in IPMNs. 

Although not extensive, these publications can be considered (listed in reverse chronological order) 

  1. https://pubmed.ncbi.nlm.nih.gov/33465354/
  2. https://pubmed.ncbi.nlm.nih.gov/31323382/
  3. https://pubmed.ncbi.nlm.nih.gov/29404638/
  4. https://pubmed.ncbi.nlm.nih.gov/31542380/

Author Response

February 8, 2021

Prof. Dr. Samuel C. Mok

Editor-in-Chief

Cancers

Manuscript ID: cancers-1093265, R1

Pre-operative imaging and pathological diagnosis of localized high-grade pancreatic intra-epithelial neoplasia without invasive carcinoma

Thank you very much for considering our manuscript and providing us with constructive comments from the Editor and Reviewers. The suggestions have greatly improved our manuscript, and we believe that it is now suitable for publication in Cancers. Point-by-point responses to the Editor’s and Reviewers’ comments are provided in the attached file.

We look forward to hearing from you.

Best regards,

Ryota Sagami, MD

Department of Gastroenterology, Oita San-ai Medical Center

Phone: +81-97-541-5218

Fax: +81-97-541-1311

Reviewer 4.

This is an excellent review highlighting the field of high-risk pancreatic lesions progressing to PDAC. The review extensively addresses multiple aspects of precursor lesions - PanIN and IPMN progressing to PDAC. While the review leans heavily into PanIN and early detection of high-grade PanINs, there is minimal detail about IPMNs with high-grade dysplasia. I recommend that the authors consider adding a paragraph to address investigations in early and/or accurate detection of high-grade dysplasia in IPMNs. 

Although not extensive, these publications can be considered (listed in reverse chronological order) 

  1. https://pubmed.ncbi.nlm.nih.gov/33465354/
  2. https://pubmed.ncbi.nlm.nih.gov/31323382/
  3. https://pubmed.ncbi.nlm.nih.gov/29404638/
  4. https://pubmed.ncbi.nlm.nih.gov/31542380/

Response: We thank the reviewer for the constructive comments. Accordingly, we added new text describing imaging of high-grade IPMN in section 3.3.1, along with the recommended references, as detailed below. We have chosen to not separate this information into a new section to maintain the focus of the manuscript.

Page 5, lines 177–179

In addition, surveillance of branch-duct IPMN should be focused on two types of carcino-genesis, carcinoma derived from IPMN; intraductal papillary mucinous carcinoma (IPMC) and concomitant PDAC (de novo PDAC) [78-82].

Page 5, lines 180–206

3.3.1. IPMN with high-grade dysplasia

IPMN exhibits multistep progression of low-grade IPMNs to high-grade IPMN and IPMC [9]. The American Gastroenterological Association and the International Associa-tion of Pancreatology have described the high-risk radiological futures of IPMN [72,83], and the resection of branch-duct IPMN is performed based on these guidelines [72,73,82]. Mural nodules may be the most predictive finding of high-grade IPMN and IPMC [83,84,85] and can be detected by existing radiological imaging modalities, such as en-hanced computed tomography (CT), magnetic resonance imaging (MRI), and endoscopic ultrasound (EUS) [73,86-88]. The diagnostic sensitivity and specificity of mural nodules 5–10 mm in diameter in the context of IPMC and high-grade IPMN are 73–100% and 73–85%, respectively [89-95]. The cutoff size for mural nodules is 5 mm or more [96], and most branch-duct IPMNs without mural nodules remain unchanged during long-term follow-up (median, 57 months) [85]. Pancreatoscopy is also useful because it enables di-rect visualization of lesions in pancreatic duct and direct biopsy [97]. In particular, this approach is useful for differential diagnosis between malignant and benign IPMN with an accuracy of 67% for branch duct IPMNs [98]. Using the technique of EUS-fine needle aspiration (EUS-FNA), cyst fluid analysis and confocal laser endoscopy can be performed [7,88]. Cyst fluid cytology has 90% specificity for the diagnosis of high-grade IPMN and IPMC [66,88,99]. In addition, to detect small IPMC or high-grade IPMN, DNA-based examination of pancreatic cyst fluid is useful [100], showing a sensitivity and specificity of 89% and 100%, respectively, which are higher than those of mural nodules (32% and 94%, respectively) and malignant cytopathology (32% and 98%, respectively) [101]. Confocal laser endoscopy identifying a vascular network pattern representing subepithelial capillary vascularization using endomicroscopy may useful for distinguishing high-grade IPMN and IPMC from IPMN without high-grade dysplasia, with 83–88% sensitivity and 88–100% specificity [102,103]. Thus, imaging and histopathological diagnostic methods for IPMC or high-grade IPMN have been established to some extent.

Page 5, lines 218–220

However, no imaging diagnostic methods have been developed to distinguish high-grade PanIN with coexisting IPMN from high-grade IPMN alone.